# Clinical Experience with Targeted Alpha-Emitter Peptide Receptor Radionuclide Therapy (α-PRRT) for Somatostatin Receptor-Positive Neuroendocrine Tumors

**DOI:** 10.3390/ph18111608

**Published:** 2025-10-24

**Authors:** Hannes Leupe, Merel Cauwenbergh, Frederik Cleeren, Jeroen Dekervel, Chris Verslype, Christophe M. Deroose

**Affiliations:** 1Nuclear Medicine, University Hospitals Leuven & Nuclear Medicine and Molecular Imaging, Department of Imaging and Pathology, University of Leuven, Herestraat 49, 3000 Leuven, Belgium; hannes.leupe@uzleuven.be (H.L.);; 2Radiopharmaceutical Research, Department of Pharmacy and Pharmacology, University of Leuven, 3000 Leuven, Belgium; 3Digestive Oncology, University Hospitals Leuven, 3000 Leuven, Belgium

**Keywords:** targeted alpha therapy, TAT, radionuclide therapy, neuroendocrine tumors, actinium-225, ^225^Ac-DOTATATE, lead-212, ^212^Pb-VMT-α-NET, peptide receptor radionuclide therapy, PRRT

## Abstract

**Background:** α-emitting Peptide Receptor Radionuclide Therapy (α-PRRT) is emerging as a promising new generation of PRRT for neuroendocrine tumors (NETs), providing enhanced tumor cell cytotoxicity and reduced irradiation of adjacent healthy tissues due to its high linear energy transfer (LET) and short particle range. This review summarizes available clinical evidence on α-PRRT using different α-emitting isotopes, including actinium-225, lead-212, and bismuth-213, in somatostatin receptor (SSTR)-positive NETs. **Methods:** A comprehensive literature search was conducted across PubMed, Embase, Cochrane Library, Scopus, Web of Science, and ClinicalTrials.gov, as well as major oncology congress abstracts (ENETS, ESMO, ASCO). Eligible studies included clinical trials evaluating α-PRRT in patients with advanced SSTR-positive NETs, reporting therapeutic response and adverse events. The primary endpoint was the objective response rate (ORR); secondary endpoints included disease control rate (DCR), progression-free survival (PFS), overall survival (OS), and safety. **Results:** Seven studies encompassing 150 patients were included. Treatment with [^225^Ac]Ac-DOTATATE yielded a pooled ORR of 50% and a DCR of 81.3% across 121 evaluable patients. The best responses were observed in patients who had previously responded to β-PRRT (ORR 70.4%, DCR 96.3%), while one-third of β-PRRT–refractory patients achieved partial or complete responses. [^212^Pb]Pb-DOTAMTATE demonstrated an ORR of 56.8% and DCR of 100% in preliminary phase II results, though dysphagia was noted in 34% of patients. [^213^Bi]Bi-DOTATOC and [^212^Pb]Pb-VMT-α-NET studies also showed promising disease control with minimal grade ≥ 3 hematologic or renal toxicities. Across all studies, α-PRRT was well tolerated, with predominantly low-grade hematologic adverse events and no significant hepatic or renal toxicity. **Conclusions:** Clinical data to date indicate that α-PRRT offers meaningful therapeutic benefit in patients with metastatic or treatment-refractory NETs, achieving favorable response rates with manageable toxicity. Early results support α-PRRT as a potential first- or second-line therapeutic option. Ongoing phase III trials will be critical to confirm its long-term safety, survival outcomes, and role in routine clinical practice.

## 1. Introduction

Neuroendocrine neoplasms (NENs) are a heterogeneous group of tumors arising from neuroendocrine cells. They can grow anywhere in the body but most commonly arise at the gastro-intestinal tract [1]. The incidence of NENs is estimated at 8.5 per 100,000 persons per year and is still increasing [2]. Because neuroendocrine cells are responsible for controlling homeostatic processes in the body, NENs can either be functional, meaning they secrete hormones, or non-functional [3]. They can be divided into well-differentiated neuroendocrine tumors (NETs) and poorly differentiated neuroendocrine carcinomas (NECs). Based on the WHO criteria, well differentiated NETs can be classified into grade 1 (G1), grade 2 (G2), and grade 3 (G3) based on their Ki-67 proliferation index and mitotic count, with G1 being the slowest and G3 being the fastest growing. NECs are poorly differentiated and more aggressive compared to NETs [4]. This review will focus solely on NETs.

An important characteristic of NETs is their overexpression of the somatostatin receptor (SSTR), most commonly subtype SSTR2 [5]. Up to 90% of gastroenteropancreatic (GEP) NETs express SSTRs on their cell membrane. This receptor overexpression on the cell surface offers opportunities for both therapy and diagnosis. Synthetic somatostatin analogs (SSAs) are used in first-line treatment of NETs and exert their function through agonist binding to the SSTR, causing internalization of the receptor followed by a decrease in hormone secretion, inhibition of proliferation, and a stimulation of apoptosis or cell death [6]. Treatment with non-radioactive SSAs forms the basis for therapy in SSTR-expressing NET patients, and clinical trials such as the CLARINET and PROMID studies have demonstrated significantly improved progression-free survival (PFS) compared to placebo [7,8].

On one hand, SSAs can be radiolabeled with positron-emitting isotopes such as gallium-68, copper-64, and fluorine-18 to enable PET imaging of SSTRs [9,10]. These radiolabeled SSAs bind to SSTRs with high affinity and are internalized by NET cells, leading to intracellular accumulation. The emitted radiation can then be detected and quantified, enabling the visualization of primary tumors and metastatic sites via molecular imaging. It is now standard practice to provide a SSTR PET/CT scan in the workup of every NET patient for staging, therapy selection, or follow-up [11]. On the other hand, SSAs can be radiolabeled with therapeutic isotopes for peptide receptor radionuclide therapy (PRRT), a systemic targeted treatment in which radiolabeled SSAs deliver cytotoxic radiation to tumors by binding to SSTRs [12]. The first generation of PRRT was developed in the early 1990s and used [^111^In]In-pentetreotide (Octreoscan^®^), but this gamma and Auger electron emitter was abandoned because of short-lived effects, poor penetration range, and kidney toxicity [13]. The second-generation used the β-emitting yttrium-90 attached to DOTATATE or DOTATOC. This β-emitter provided a more potent dose delivery compared to [^111^In]In-pentetreotide [14]. This β-PRRT resulted in higher efficacy, with better objective responses and longer PFS compared to the first generation PRRT [15]. However, the longer β-particle path length led to higher absorbed doses to the radiosensitive glomeruli, resulting in significant permanent kidney damage [16]. In a study of 1109 patients treated with [^90^Y]Y-DOTATOC, 9.2% of patients developed grade 4 to 5 permanent renal toxicity [17].

The third-generation radionuclide, lutetium-177, is a dual-emitting isotope that releases both β-particles and γ-photons. It is commonly conjugated with the somatostatin analog DOTATATE, forming the therapeutic radiopharmaceutical [^177^Lu]Lu-DOTATATE. It is typically administered as four cycles of 7.4 GBq each, with 8 ± 1 weeks between each cycle. Both the NETTER-1 and NETTER-2 clinical studies compared the use of [^177^Lu]Lu-DOTATATE and long-acting octreotide as treatment of NETs against a control treatment of high-dose long-acting octreotide in locally advanced or metastatic midgut G1-G2 NETs and G2-G3 gastroenteropancreatic (GEP) NETs, respectively [18,19]. The NETTER-1 trial demonstrated a significantly longer PFS (65.2% vs. 10.8% at 20 months) and higher objective response rate (ORR) (18% vs. 3%) in the β-PRRT-treated group. Additionally, a nearly 12-month difference in median overall survival (OS) was observed, favoring the β-PRRT group. However, after five years of follow-up, the final OS difference was not statistically significant, in part due to substantial cross-over towards PRRT in the control group. Another reported endpoint, health-related quality of life (HRQoL), assessed by time to QoL deterioration, was significantly improved in the β-PRRT group [18,20,21]. Similarly, the NETTER-2 trial reported a higher objective response rate and a 14-month difference in median PFS between the two groups in favor of β-PRRT in first-line setting, suggesting that PRRT could have a place in the first-line setting for patients with grade 2 or 3 GEP-NETs [19]. Following the results of the NETTER-1 trial, the European Medicines Agency (EMA) and US Food and Drug Administration (FDA) approved [^177^Lu]Lu-DOTATATE as a second-line systemic therapy in GEP-NET patients with positive SSTR expression in 2017 and 2018, respectively [22].

Despite currently being one of the more effective therapeutic options for NETs, PRRT with β-emitting radionuclides such as yttrium-90 and lutetium-177 poses some challenges. The relatively long tissue range of 2–12 mm of β-emitters implies that, after accumulation in the tumor cells, adjacent healthy surrounding tissue is also irradiated. This leads to higher risk of damage to kidneys and bone marrow, which are clinically relevant adverse events (AE) [23]. Compared to [^90^Y]Y-DOTATOC, renal toxicity with [^177^Lu]Lu-DOTATATE is rare, with several studies showing renal toxicity in less than 1% of PRRT patients [24,25]. Nonetheless, up to 4% of patients develop long-term myelodysplasia, persistent cytopenia, or leukemia [26]. Furthermore, ^177^Lu-PRRT mostly leads to resolution of symptoms and disease stabilization and is not curative [18]. Disease control is often temporary and patients often develop resistance to ^177^Lu-PRRT. It has been shown that most patients who achieve disease stabilization relapse within 2–3 years of starting ^177^Lu-PRRT [27].

Therefore, there is an increased interest in using α-emitting radionuclides for PRRT. α-particles consist of 2 protons and 2 neutrons, i.e., a helium-4 nucleus. Due to this higher mass, α-emitters have a shorter tissue range (40–100 µm) compared to β emitters (2–12 mm) [28]. As a consequence, less irradiation is delivered to healthy tissue and organs. Additionally, α-radiation has a higher linear energy transfer (LET), which is a term used to measure the molecular damage of a particle per unit length. This higher LET (80–100 keV/µm vs. 0.2 keV/µm for α vs. β particles, respectively) results in more complex and severe cell damage than low-LET damage, such as DNA double-strand breaks. These complex DNA double-strand breaks are more difficult to repair, leading to enhanced tumor cell cytotoxicity [29]. These characteristics are a significant advantage compared to β-emitters, and α-PRRT has the potential to overcome refractory disease and improve clinical outcomes [30].

This review aims to give an extensive overview of the initial clinical experiences using targeted alpha therapy (TAT) with actinium-225, lead-212, and bismuth-213 in NETs. With half-lifes of 9.9 days, 10.6 h, and 45.6 min, respectively, they are well-suited for forming stable complexes with chelator-conjugated peptides, enabling effective targeted therapy [30,31].

## 2. Methods

### Search Strategy

An extensive search of the Cochrane Central Library, Embase, PubMed, Scopus, Web of Science, and clinicaltrials.gov was performed without narrowing filters on publication date. The search concepts were “bismuth-213” or “actinium-225” or “astatine-211” or “lead-212” or “212Pb” or “213Bi” or “211At” or “225Ac” or “213Bi-DOTATATE” or “225Ac-DOTATATE” or “212Pb-DOTAMTATE” combined with “neuroendocrine tumor” or “targeted alpha therapy” or “somatostatin”. Congress abstracts of the yearly American Society of Clinical Oncology (ASCO), European Neuroendocrine Tumor Society (ENETS), and European society for Medical Oncology (ESMO) congresses were also screened. Inclusion criteria for eligibility were as follows: a peer-reviewed article or congress abstract on α-PRRT in patients with an advanced NET and a baseline SST PET/CT showing significant SST expression, reporting, respectively, on therapy response and adverse events (AE). Exclusion criteria were trials with a study population of less than five patients and studies using tandem therapy or α-PRRT with radionuclides other than actinium-225, astatine-211, lead-212, or bismuth-213. References of the selected articles were also cross-checked to ensure comprehensive coverage of relevant studies.

## 3. Results

### 3.1. Population Characteristics

In total, seven studies with 150 participants were identified (the 2020 study from Ballal. et al. was not counted due to overlap in patient population with their more recent publication [32]). Baseline characteristics (Table 1 and Table 2) varied widely across studies, including differences in design, sample size (7–91 patients), age (23–84 years), performance status, primary tumor sites, metastatic spread, treatment regimens, and follow-up duration (2–41 months) [32,33,34,35,36,37,38,39]. All 150 patients underwent a baseline [^68^Ga]Ga-DOTATATE or [^68^Ga]Ga-DOTATOC PET-CT followed by subsequent targeted α-therapy and follow-up scans at study-specific intervals. Each study administered [^225^Ac]Ac-DOTATATE, [^212^Pb]Pb-DOTAMTATE, or [^213^Bi]Bi-DOTATOC at eight-week or two-month intervals [33,34,35,36,37,38]. [^212^Pb]Pb-VMT-α-NET was administered as a single dose [39]. The specifics of the regimen, such as the number of cycles and cumulative activity, varied among the studies and are summarized in Table 1. Morphological assessment was performed with RECIST 1.1 criteria, unless otherwise specified [40]. An overview of the clinical outcomes is shown in Table 3. Ongoing clinical trials are shown in Table 4.

### 3.2. Efficacy and Safety of [^213^Bi]Bi-DOTATOC

The Heidelberg group was the first to publish clinical results on targeted alpha therapy (TAT) using intra-arterial injection of [^213^Bi]Bi-DOTATOC in eight patients with progressive, advanced metastatic NETs refractory to previous [^90^Y]Y-DOTATOC or [^177^Lu]Lu-DOTATOC treatment [36]. Of these, seven patients were treated for liver-dominant disease with a median of five cycles (range: 2–5) intra-arterial administration of [^213^Bi]Bi-DOTATOC via the hepatic artery, while one patient had diffusely disseminated bone marrow metastases and was treated with one cycle of systemic [^213^Bi]Bi-DOTATOC via intravenous administration. At a follow-up interval of 12–34 months, treatment with intra-arterial [^213^Bi]Bi-DOTATOC in seven patients resulted in one complete response (CR), two partial responses (PR), and stable disease (SD) in three patients (Figure 1). One patient had radiological response in hepatic lesions, without response assessment of skeletal metastases. Two patients later developed extrahepatic progression at 12 and 15 months, despite stable liver lesions. The patient with diffuse bone marrow involvement receiving one cycle of systemic α-PRRT had satisfactory bone lesion targeting with no significant bone marrow toxicity.

The acute hematological toxicity of [^213^Bi]Bi-DOTATOC therapy was minimal in the majority of patients. Two patients exhibited chronic grade 1 or 2 anemia and one patient with a history of grade 4 thrombocytopenia developed temporary grade 2 thrombocytopenia. Notably, one patient developed a myelodysplastic syndrome (MDS), which then progressed to acute myeloid leukemia, ultimately leading to the patient’s death. This patient was heavily pre-treated with multiple cycles of [^90^Y]Y-DOTATOC and [^177^Lu]Lu-DOTATOC. Acute renal toxicity was moderate, with a decrease in mean glomerular filtration rate (GFR) of 30%. No significant liver toxicity or major clinical AE were reported within this cohort [36].

Despite these encouraging results, the widespread clinical use of [^213^Bi]Bi-DOTATOC therapy was hampered by logistical challenges related to the short half-life of bismuth-213 (45.6 min). This short half-live restricts transport to external facilities, as the narrow window between isotope production and patient administration necessitates on-site production [30,31]. Automated synthesis modules directly coupled to ^225^Ac/^213^Bi generators may help streamline the on-site production process, but widespread implementation also depends on the adequate availability of ^225^Ac to supply the ^225^Ac/^213^Bi generators and meet clinical demand.

### 3.3. Efficacy and Safety of [^225^Ac]Ac-DOTATOC/[^225^Ac]Ac-DOTATATE

The logistical issues regarding the short half-life of bismuth-213 prompted researchers to investigate [^225^Ac]Ac-DOTATOC as a more viable alternative. With a significantly longer half-life of 9.9 days, distribution of ^225^Ac becomes feasible to allow for clinical distribution without the necessity for in-house production. In a small pilot study involving ten patients with metastatic NENs refractory to previous beta-PRRT, intra-arterial [^225^Ac]Ac-DOTATOC was found to be both well tolerated and therapeutically effective [41]. Intra-arterial delivery of the radiopharmaceutical could theoretically improve tumor dose while minimizing systemic toxicity. However, as with earlier intra-arterial approaches using bismuth-213, this method faces challenges such as technical complexity and a lack of widespread interventional radiology expertise in administering PRRT agents [42].

These limitations of intra-arterial alfa targeted therapy have led to the interest in systemic alfa therapy. The Heidelberg group conducted an initial dose-escalation study for systemic [^225^Ac]Ac-DOTATOC treatment administered intravenously, both for a single cycle and for fractionated regimens. They determined that a single cycle was well tolerated at a maximum dose of 40 MBq, while fractionated doses administered every 8 to 12 weeks (up to a cumulative total of 75 MBq) were also tolerated without causing delayed toxicity [43]. Subsequently, they published a retrospective study evaluating [^225^Ac]Ac-DOTATOC in 39 NET patients [44]. Out of these, 33 patients (86.5%) had previously received PRRT using beta- or alpha-emitters such as yttrium-90, lutetium-177, or bismuth-213 before undergoing treatment with [^225^Ac]Ac-DOTATOC. Despite this prior intensive treatment, no grade ≥ 3 hematologic toxicities occurred, even with cumulative injected activities up to 60–80 MBq. Chronic renal toxicity was observed in 2 patients, but pre-existing renal risk factors were important comorbidities and no grade 3 or higher AE were observed.

The use of intravenously administered [^225^Ac]Ac-DOTATATE was first described by Ballal et al. in 2020 [32]. Initial findings from a cohort of 32 NET patients previously treated with ^177^Lu-PRRT demonstrated that [^225^Ac]Ac-DOTATATE treatment every eight weeks was effective and induced sustained responses. Among the 24 patients who underwent interim morphological assessment, 15 (62.5%) achieved a PR, and the other 9 patients (37.5%) achieved SD (Figure 2). Notably, no cases of PD were reported, and no patient deaths occurred during a short median follow-up period of 8 months (range: 2–13 months). Treatment-related toxicities were minimal, transient, and non-life-threatening. They later reported the long-term follow-up data in an expanded cohort of 91 GEP-NET patients, one of the largest cohorts of patients treated with α-PRRT so far [33]. This prospective study was conducted in a real-world setting based on everyday clinical practice and included patients that were often pre-treated (62.6% received prior ^177^Lu-PRRT) or had poor performance status (31% had ECOG status ≥ 3). They were treated with a median of four cycles intravenously administered [^225^Ac]Ac-DOTATATE at 100 kBq/kg with mean cumulative administered activity of 35.5 MBq (range: 21.6–59.5 MBq) and combined with oral capecitabine, a radiosensitizing oral pro-drug of 5-fluorouracil, as a radiosensitizer during each cycle. At a median follow-up period of 24 months (range 5–41 months), median OS and PFS were not attained and 24-month PFS and OS probabilities were 67.5% and 70.8%, respectively. Response evaluation was performed with RECIST 1.1 criteria. Of the 79 evaluable patients, 2 (2.5%) achieved CR, 38 patients (48.1%) PR, 23 patients SD (29.1%), and 16 patients (20.3%) PD, corresponding to an ORR of 50.6% and DCR of 79.7%. In addition to morphological response, the median Karnofsky performance score improved from 60 pre-treatment to 70 post-treatment, indicating better overall well-being. Demirci et al. retrospectively evaluated [^225^Ac]Ac-DOTATATE treatment in 11 metastatic NET patients with different primary sites. Of these, 10 (90.9%) received prior ^177^Lu-PRRT. Mean administered activity was 8.2 ± 0.6 MBq per cycle with a median of one cycle (range 1–3) per patient. Median follow-up time was not specifically mentioned in the paper. Response assessment with RECIST 1.1 in 9 patients showed 1 patient (11.1%) with PD, 4 (44.4%) with SD, and 4 (44.4%) with PR, corresponding to a DCR of 88.9% [35].

Extending the use of [^225^Ac]Ac-DOTATATE beyond GEP-NETs, Yadav et al. investigated the use of [^225^Ac]Ac-DOTATATE in advanced-stage paragangliomas (PGLs) in a pilot study with nine patients [38]. They were treated with a median of 3 cycles (range: 2–9) [^225^Ac]Ac-DOTATATE (100 kBq/kg) at 8-week intervals up to a mean cumulative activity of 42.2 ± 27 MBq, combined with oral capecitabine. Following treatment with [^225^Ac]Ac-DOTATATE, response assessment in eight patients, at a median follow-up duration of 22.5 months, showed a PR in four (50.0%) patients, SD in three (37.5%), and PD in one (12.5%), resulting in a DCR of 87.5% (Figure 3). Additionally, improvements were seen in functional performance metrics and clinical symptoms. In seven patients treated for antihypertension, antihypertensive mediation was completely stopped in two and reduced in three patients [38]. Yang et al. further evaluated the use of [^225^Ac]Ac-DOTATATE in a NEN cohort of 10 patients predominantly including PGL (n = 4) and pheochromocytoma (PCC) (n = 3) [37]. They received a median of 3 cycles (range: 2–6) at 100 kBq/kg with a mean cumulative activity of 22.9 ± 9 MBq. Notably, response evaluation was performed according to PERCIST 1.0 criteria (slightly modified for SST PET) at a median follow-up time of 14 months (range: 7–22) [45]. Among these patients, four (40%) had PR, four (40%) SD, and two (20%) PD, corresponding with an ORR of 40.0% and a DCR of 80.0% [37]. These studies show that [^225^Ac]Ac-DOTATATE can also be an effective therapeutic option in PGL and PCC patients.

In total, these four studies evaluating [^225^Ac]Ac-DOTATATE included 121 patients [33,35,37,38], not taking into account pilot study from Ballal et al. due to overlap in patient population [32]. Of these, three studies used RECIST 1.1 criteria to assess response evaluation in a total of 96 patients. Of the 96 patients, 2 (2%) achieved a CR, 46 (48%) had a PR, 30 (31%) achieved SD, and PD was observed in 18 (19%) patients, corresponding to a pooled ORR of 50.0% (48/96 patients) and pooled DCR of 81.3% (78/96) [33,35,38].

Based on prior therapy, the patients in the [^225^Ac]Ac-DOTATATE α-PRRT arm can be categorized into three subgroups: a ^177^Lu-PRRT naïve group, a ^177^Lu-PRRT refractory group (defined as progressed during or within 12 months of completion of ^177^Lu-PRRT treatment), and a ^177^Lu-PRRT responding group. This subdivision was made in three of the four studies, and response assessment categories were reported for 97 of them [33,37,38]. Patients previously responsive to ^177^Lu-PRRT (n = 27) demonstrated the best outcomes (pooled ORR and DCR of 70.4% and 96.3%, respectively), followed by PRRT-naïve (n = 32) patients (pooled ORR and DCR of 53.1% and 87.5%, respectively), while ^177^Lu-PRRT refractory patients (n = 38) had the lowest pooled ORR and DCR of 31.6% and 64.8%, respectively, which is still a good outcome in pretreated, progressive NET patients. Moreover, Ballal et al. reported survival outcomes on the previously mentioned subgroups [33]. At 24 months, the estimated OS probability was 62.6% in the ^177^Lu-PRRT-naïve subgroup whereas this was 95.0% among patients who had previously responded to β-PRRT. In contrast, the refractory subgroup demonstrated the lowest 24-month OS probability of 55.6% [33], with this subgroup being the only cohort in which median OS was reached, measured at 26 months. Demirci et al. did not distinguish between ^177^Lu-PRRT naïve patients and those who had received prior ^177^Lu-PRRT in their reported outcomes [35].

An overview of AEs following the administration of [^225^Ac]Ac-DOTATATE is shown in Table 5. Overall, treatment was well tolerated with transient low-grade hematologic AEs, such as low grade thrombocytopenia or anemia, being the most common side effect. There were minimal kidney and liver function toxicities. Only one patient experienced transient grade 3 thrombocytopenia. None of the patients experienced AEs severe enough to discontinue treatment [33,35,37,38].

Ongoing clinical phase I trials, such as NCT06732505 are currently further evaluating safety and efficacy of [^225^Ac]Ac-DOTATATE in GEP-NET patients. The ACTION-1 trial (NCT05477576) is a phase I/III study comparing 4 cycles 120 kBq/kg [^225^Ac]Ac-DOTATATE (RYZ101) with the standard of care in SST-expressing GEP-NET patients progressive after ^177^Lu-PRRT in its phase III part. In an interim safety analysis of nine patients treated with 4 cycles at a starting injected activity of 120 kBq/kg, no activity-limiting toxicity was seen [46].

### 3.4. Efficacy and Safety of [^212^Pb]Pb-DOTAMTATE/[^212^Pb]Pb-VMT-α-NET

A collaboration between Radiomedix and OranoMed has led to the development of [^212^Pb]Pb-DOTAMTATE (AlphaMedix™). In a phase I first-in-human dose-escalation trial, Delpassand et al. investigated efficacy and safety of [^212^Pb]Pb-DOTAMTATE in 20 histologically confirmed NET patients [34]. The included patients had no prior history of β- or α-PRRT. The treatment regimen consisted of four cycles of 2.5 MBq/kg [^212^Pb]Pb-DOTAMTATE at 8-week intervals, administered intravenously. Mean cumulative administered activity over four cycles was 791 MBq (range: 681–873 MBq). After [^212^Pb]Pb-DOTAMTATE therapy, one patient achieved a CR, seven had PR, and two had SD according to the RECIST 1.1 criteria at a median follow-up time of 17.4 months (range: 9–26 months) (Figure 4). ORR and DCR were 80% and 100%. Furthermore, a median duration of response of 14 months and a median time to response of 5.2 months was reported. Ten patients receiving the highest injected activity of [^212^Pb]Pb-DOTAMTATE experienced no grade 4 adverse events. The severity of most reported adverse events was grade 1 or 2. No significant hematological toxicity was reported, except for a temporary decrease in mean lymphocyte count, which reverted within 6 months post treatment. One patient developed acute kidney injury two weeks after the last treatment cycle, resulting in persistent chronic kidney disease. However, this patient had several confounding factors and the investigators deemed the kidney injury unrelated to the [^212^Pb]Pb-DOTAMTATE treatment, although a contribution of the α-PRRT cannot be excluded [34]. Based on these promising results, the American Food and Drug Administration (FDA) has granted [^212^Pb]Pb-DOTAMTATE ‘Breakthrough Therapy Designation’, signifying the drug’s potential to significantly improve over existing treatments [47]. An ongoing phase II multicenter trial (ALPHAMEDIX-02; NCT05153772) is evaluating the efficacy and safety of [^212^Pb]Pb-DOTAMTATE in 36 PRRT-naïve and 30 PRRT-refractory patients NETs. Preliminary results from the PRRT-naïve cohort, presented at the ASCO 2024 Annual Meeting [48], demonstrated an ORR of 55.6% (20/36), with a median response duration of 17 months. The combined ORR across both phase I [34] and phase II trials was 56.8%. At 24 months, the probabilities of PFS and OS were 74.3% and 90.7%, respectively. High-grade hematologic toxicity was uncommon and limited to reversible grade 3 or 4 lymphocytopenia. Dysphagia occurred in 34% (15/44) of patients, with one case of grade 3/4 severity presenting as an “achalasia-like” syndrome that responded to botulinum toxin (Botox) injections [48].

VMT-α-NET is another novel SSTR2 targeting peptide that can be radiolabeled with both ^212^Pb, an alfa emitter, or ^203^Pb, a gamma ray emitter [39]. Michler et al. evaluated this promising theranostic isotope pair in 12 progressive, metastatic NET patients. This cohort was heavily pretreated, with all patients having received ^177^Lu-PRRT and 50% of patients having received [^225^Ac]Ac-DOTATATE therapy prior to treatment with [^212^Pb]Pb-VMT-α-NET. Of these 12 patients, 2 died due to disease progression before receiving the treatment and another 2 patients were excluded from therapy based on poor uptake on pretherapeutic imaging with [^203^Pb]Pb-VMT-α-NET, resulting in 8 treated patients. Treatment consisted of a single dose of [^212^Pb]Pb-VMT-α-NET of 1.2 MBq/kg. Post-therapeutic [^212^Pb]Pb-VMT-α-NET SPECT/CT imaging showed comparable, but lesion-dependent heterogeneous tumor and normal tissue comparable to pretreatment [^203^Pb]Pb-VMT-α-NET SPECT imaging. Response evaluation using RECIST 1.1 showed SD in all patients (8/8) 3 months after treatment (Figure 5). Treatment was well tolerated by all patients treated, and no hepatic or renal toxicity was noted. Hematological toxicity was limited, with only a reported decrease in absolute lymphocyte count to three months after therapy [39]. In an ongoing phase I/IIa, first-in-human clinical trial (NCT05636618), the safety and preliminary efficacy of four cycles of [^212^Pb]Pb-VMT-α-NET were evaluated in PRRT-naïve NET patients. Updated results were presented at the 2025 ASCO Annual Meeting [49]. Efficacy analysis included nine patients: seven received four cycles of 185 MBq, of whom one exhibited PD, two had SD, and four achieved PR; two patients treated with four cycles of 92.5 MBq both demonstrated SD. The ORR and DCR across both dosing cohorts were 4/9 (44.4%) and 8/9 (88.9%), respectively. Safety was assessed in all treated patients (n = 42) with a median follow-up of 14 months. No grade 4 or higher AEs or cases of renal insufficiency were observed [49]. Phase I and phase II studies are currently being conducted to confirm these results in larger cohorts with longer follow-up (NCT06479811, NCT06427798, NCT06148636).

## 4. Discussion

Clinically, α-PRRT or TAT represents a highly promising approach for the treatment of SSTR-positive NETs. Compared with conventional β-PRRT, α-particle PRRT offers distinct advantages owing to their high LET, causing more complex double-strand DNA breaks that are difficult to repair.

Emerging clinical data suggest that TAT can achieve higher objective response rates than β-emitters. Initial experience with [^213^Bi]Bi-DOTATOC, delivered via intra-arterial hepatic administration, demonstrated encouraging results in heavily pretreated, progressive metastatic NET patients. However, widespread adoption of [^213^Bi]Bi-DOTATOC is hindered by its short half-life (45.6 min) and limited availability of ^225^Ac, which necessitates on-site production and restricts broader clinical use.

The development of [^225^Ac]Ac-DOTATOC and [^225^Ac]Ac-DOTATATE, with a longer half-life (9.9 days), addressed these logistical limitations and marked a transition to systemic administration. Multiple studies, including a large prospective cohort of 91 GEP-NET patients, demonstrated that intravenous [^225^Ac]Ac-DOTATATE achieved a pooled ORR of 50.0% (48/96 patients) and pooled DCR of 81.3% (78/96), even in patients heavily pretreated with β-PRRT [33,35,38]. In the available scientific literature, only one study evaluating [^225^Ac]Ac-DOTATATE reported a median PFS, which was 12 months, and it was observed in a cohort where 10 of 11 patients were refractory to prior [^177^Lu]Lu-DOTATATE therapy [35]. In all other studies, median PFS was not reached, and similarly, no trial has yet reported a median OS. For comparison, initial findings from the NETTER-1 and NETTER-2 trials with [^177^Lu]Lu-DOTATATE also demonstrated high objective response rates (ORR) and prolonged PFS, but the final analysis revealed no statistically significant difference in OS for patients in NETTER-1 [19,21]. Collectively, these early data suggest that α-PRRT has the potential to improve both median PFS and OS, particularly in treatment-refractory populations. However, longer-term follow-up and confirmatory prospective phase 3 trials, such as the Action-1 (NCT05477576), are necessary to definitively assess the survival benefit of [^225^Ac]Ac-DOTATATE.

Pooled subgroup analyses revealed the greatest benefit in patients previously responding to ^177^Lu-PRRT (ORR 70%, DCR 96%), followed by PRRT-naïve patients (ORR 53%, DCR 88%), whereas refractory patients showed lower responses (ORR 32%, DCR 65%). The latter still represent clinically meaningful outcomes in patients that have exhausted one of the most potent NET therapies, ^177^Lu-PRRT. Survival data, though limited, indicated promising 24-month OS probabilities of 95% and 63% in the responding and PRRT-naïve subgroups, respectively, compared to 56% in refractory patients. One potential explanation is the variation in baseline tumor biology and patient characteristics present prior to [^225^Ac]Ac-DOTATATE therapy. Patients who previously responded to ^177^Lu-PRRT may have more indolent disease biology, higher SSTR expression, or slower tumor proliferation rates, making them more responsive to subsequent [^225^Ac]Ac-DOTATATE therapy. In contrast, refractory patients may have more aggressive disease, heterogeneous or reduced SSTR expression, or underlying molecular resistance mechanisms that limit α-particle efficacy. However, [^225^Ac]Ac-DOTATATE demonstrates clinically meaningful efficacy even in patients refractory to β-emitting PRRT, suggesting that α-PRRT can still overcome some mechanisms of β-PRRT resistance. This supports the use of [^225^Ac]Ac-DOTATATE as a salvage therapy in patients who have exhausted β-PRRT options.

Novel SSAs labeled with α-emitters, such as [^212^Pb]Pb-DOTAMTATE, have also demonstrated promising efficacy: In a phase I trial, [^212^Pb]Pb-DOTAMTATE demonstrated an objective response rate (ORR) of 80% and a disease control rate (DCR) of 100% in PRRT-naïve patients, with durable responses and manageable toxicity [34]. Preliminary data from an ongoing phase II study indicate sustained responses, with a pooled ORR of 56.8%, although dysphagia has been observed in approximately one-third of patients, including isolated cases of grade 3/4 severity [48]. Another α-emitting agent, [^212^Pb]Pb-VMT-α-NET, achieved disease stabilization in a phase I study of heavily pretreated patients, including those with prior ^177^Lu- and ^225^Ac-based PRRT [34]. Early results from an ongoing phase I/IIa trial in PRRT-naïve patients show an ORR of 44.4% and a DCR of 88.9%, with a favorable safety profile [49].

Across studies, α-PRRT was well tolerated by almost all patients. All studies administered the treatment simultaneously with renoprotective amino acids. Hematologic toxicity was predominantly low-grade and transient, with only isolated grade 3 thrombocytopenia reported [33]. Acute renal and hepatic toxicities were minimal, although early studies with [^213^Bi]Bi-DOTATOC observed a mean glomerular filtration rate (GFR) reduction of 30%. One case of myelodysplastic syndrome (MDS) progressing to acute myeloid leukemia occurred in a heavily pretreated patient, making a causal link with α-PRRT uncertain [36]. Importantly, no treatment discontinuations due to severe adverse events were reported for ^225^Ac- or ^212^Pb-based therapies. Unlike prostate cancer α radionuclide therapy ([^225^Ac]Ac-PSMA-617), xerostomia is rare in NET patients due to differing biodistribution, with much lower salivary gland uptake seen with the radiolabeled SSA in comparison to urea motif-based radiolabeled ligands for the prostate specific membrane antigen [50]. However, one patient developed Graves’ disease, potentially due to the expression of SSTR2 in the thyroid gland, indicating it may be an organ at risk during PRRT. It remains unclear how alpha irradiation leads to the development of thyroid hormone receptor-stimulating autoantibodies, the causative mechanism of Graves’ disease [51]. Other acute clinical adverse events, including nausea, vomiting, abdominal pain and distention, diarrhea, gastritis, fatigue, musculoskeletal pain, loss of appetite, headache, flushing, and myalgia, were reported but remained transient and comparable to adverse events seen in β-PRRT. However, the long-term safety profile, particularly regarding renal impairment and late hematologic toxicity, remains insufficiently characterized due to limited follow-up, comorbidities, and heavy pre-treatment in existing studies. The majority of patients included in this review received prior β-PRRT, which introduces difficulty in attributing an adverse event to α-PRRT (acute AEs) or β-PRRT (long-term AEs). Previous studies with PRRT (especially with yttrium-90) proved the need for continued vigilance regarding the long-term safety profile [52].

Thus far, no major drug regulating agency has approved an α-emitting radiopharmaceutical for tumor-targeted alpha therapy. The only α-emitting radiopharmaceutical with FDA and EMA approval, [^223^Ra]RaCl_2_ (Xofigo^®^), is indicated for the treatment of symptomatic bone metastases in patients with castration-resistant prostate cancer (mCRPC) [53]. [^225^Ac]Ac-PSMA-617 is another promising α-emitting radiopharmaceutical currently undergoing clinical trials for the treatment of mCRPC. However, more evidence regarding efficacy and toxicity is needed before implementing the therapy as a standard treatment [54]. Beyond NETs, α-PRRT also holds substantial potential for expanding therapeutic options in other SSTR-expressing malignancies. Due to its higher linear energy transfer and short path length, α-PRRT may effectively target tumors with lower SSTR expression or faster proliferation rates, where conventional β-emitters are less effective. Preclinical and early clinical investigations are exploring its use in small-cell lung cancer, multiple myeloma, and breast cancer. These studies highlight the potential of α-PRRT to broaden its therapeutic application across diverse SSTR-expressing tumors [55,56,57].

Current α-PRRT data for NET patients, while promising, are limited by small, heterogeneous cohorts; varying treatment regimens; and mostly retrospective studies, and in the case of prospective studies, single-arm designs. Additionally, follow-up durations remain insufficient to fully assess long-term safety, particularly hematologic and renal toxicities.

Among the α-emitters investigated for peptide receptor radionuclide therapy (α-PRRT), actinium-225 and lead-212 have emerged as the most promising candidates due to their favorable decay characteristics and compatibility with peptide-based targeting vectors.

Actinium-225 is currently the most clinically advanced α-emitter. With a half-life of 9.9 days, it emits four α-particles per decay chain, enabling highly potent tumor irradiation. Its labeling chemistry with DOTA-based peptides allows stable binding and effective systemic delivery. Clinical studies using [^225^Ac]Ac-DOTATATE have demonstrated strong therapeutic efficacy with manageable toxicity. However, a key safety concern is the recoil-induced release of radioactive daughters. The α-decay of actinium-225 generates radionuclides such as francium-221 and bismuth-213, which can detach from the chelator and redistribute to non-target tissues. Free francium-221 tends to accumulate in kidneys, salivary glands, and intestines, while bismuth-213 shows preferential retention in kidneys and liver. The magnitude of potential off-target toxicity is vector-dependent and warrants careful consideration in radiopharmaceutical design.

Lead-212, with a 10.6 h half-life, decays in vivo to Bismuth-212, thereby extending the effective therapeutic window for α-particle emission. Both local and centralized production of ^212^Pb-radiopharmaceuticals are feasible. Preclinical and early clinical evaluations of [^212^Pb]Pb-DOTAMTATE and related compounds have shown encouraging results, demonstrating the practical feasibility and scalability of this isotope. Furthermore, imaging surrogates such as lead-203 allow for dosimetry assessments and theranostic applications.

In contrast, bismuth-213 has a short half-life of 45 min, leading to rapid and intense dose deposition over a brief period. While this property can be advantageous for highly localized therapies, it limits the feasibility of systemic applications unless used with a rapidly clearing vector that minimizes renal retention. The short half-life also necessitates on-site production and quality control of ^213^Bi-labeled radiopharmaceuticals, posing significant logistical challenges. Although high-activity ^225^Ac/^213^Bi generator systems have been developed [58,59], the primary bottleneck remains the limited global availability of the parent isotope actinium-225. Ongoing efforts to expand actinium-225 production are expected to alleviate these constraints in the coming years, potentially paving the way for broader clinical adoption of ^213^Bi-based radiopharmaceuticals [31]. Global ^225^Ac production by 2032 is estimated at well above 25 TBq per year, which is sufficient to produce at least 2 million patient doses, assuming average treatment needs (average 30–36 MBq per patient) [60].

Astatine-211 possesses nearly ideal properties for α-PRRT. Its 7.2 h half-life, pure α-emission, and absence of long-lived α- or high-energy γ-emitting daughters make it particularly suitable for small peptides and rapidly clearing biomolecules. However, the chemical instability (in vivo deastatination) of many ^211^At-labeled radiopharmaceuticals remains a key challenge that must be addressed to fully harness the therapeutic potential of this isotope. Despite its favorable radiophysical and dosimetric profile, clinical translation is constrained by the restricted availability of suitable cyclotrons and radiochemistry facilities capable of producing ^211^At-radiopharmaceuticals at clinically relevant scales. Moreover, although astatine-211 itself decays relatively quickly, a small fraction of decays produce bismuth-207, with a long half-life of 31.6 years, which necessitates long-term storage or regulated disposal of waste. While the absolute activity of bismuth-207 is low, its presence imposes waste-management considerations beyond those associated with shorter-lived α-emitters such as lead-212 or bismuth-213. To date, no clinical α-PRRT studies in patients have been reported using astatine-211.

Moving forward, robust activity finding studies, with preferentially dosimetric substudies, are essential to optimize dosing schedules, establish toxicity thresholds, and define patient selection criteria, with ensuing multicenter randomized controlled trials to provide definitive efficacy and safety data. The ongoing ACTION-1 phase III trial and other early-phase studies (e.g., NCT06732505, NCT05153772) are expected to clarify the role of α-PRRT as salvage therapy and potentially as first-line treatment in SSTR-positive NETs. Integration of molecular imaging to stratify responders and combination approaches with radiosensitizers or immunotherapy may further enhance therapeutic outcomes. Lastly, SSAs labeled with the promising alpha emitter astatine-211 (half-life: 7.2 h) represent a potential future avenue for α-PRRT of NETS, although to date only preclinical studies have been reported [56].

## 5. Conclusions

α-PRRT shows strong potential as an effective treatment for SSTR-positive NENs, including metastatic and treatment-resistant cases. Early studies report encouraging response rates and manageable acute toxicity, with α-particles offering biological advantages over β-emitters. However, long-term safety, renal effects, and survival benefits remain uncertain, underscoring the need for phase III trials and extended follow-up to define the role of α-PRRT in NEN management.

## Figures and Tables

**Figure 1 pharmaceuticals-18-01608-f001:**
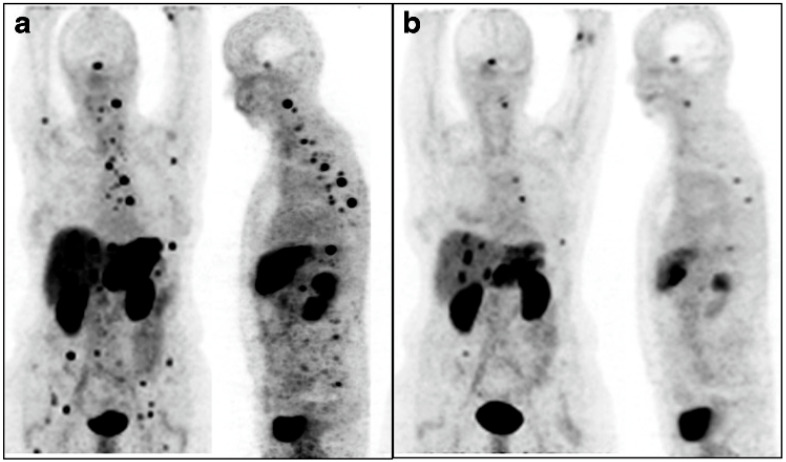
(**a**) A NET patient with an extensive tumor burden in the left liver lobe and multiple lesions in the right lobe, as well as disseminated bone marrow metastases predominantly in the spine and pelvis. These are demonstrated in coronal and sagittal ^68^Ga-DOTATOC-PET maximum intensity projections. (**b**) Liver metastases showed significant shrinkage after administration of 10.5 GBq of ^213^Bi-DOTATOC into the common hepatic artery. Additional systemic efficiency resulting from the ^213^Bi-DOTATOC reaching the systemic circulation after the first pass of the liver was noted after 6 months in that most of the bone marrow metastases had also diminished. *Figure 213. Bi-DOTATOC receptor-targeted alpha-radionuclide therapy induces remission in neuroendocrine tumors refractory to beta radiation, Eur. J. Nucl. Med. Mol. Imaging 41:2106–2119 (2014) [36], under CC BY 4.0 (https://creativecommons.org/licenses/by/4.0/). Accessed on 12 September 2025.*

**Figure 2 pharmaceuticals-18-01608-f002:**
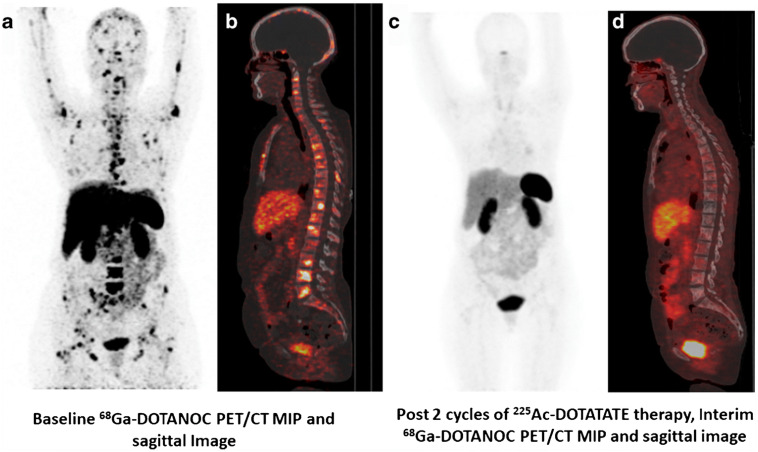
(**a**) MIP image and (**b**) fused sagittal image of a 54-year-old female diagnosed with rectal NET who had undergone 12 injections of sandostatin LAR as the first-line treatment and demonstrated disease progression after four cycles of ^177^Lu-DOTATATE PRRT and capecitabine. The baseline pre-therapy diagnostic ^68^Ga-DOTANOC PET/CT scan demonstrated somatostatin receptor (SSTR) avid extensive skeletal metastases. (**c**) MIP image and (**d**) fused sagittal image post two cycles of ^225^Ac-DOTATATE therapy, the interim diagnostic ^68^Ga-DOTANOC PET/CT scan showed partial morphological response and molecular response. *Figure 225. Ac-DOTATATE targeted alpha therapy for gastroenteropancreatic neuroendocrine tumor patients stable or refractory to ^177^Lu-DOTATATE PRRT: first clinical experience on the efficacy and safety, European Journal of Nuclear Medicine and Molecular Imaging* [32]*, © 2020 Springer Nature, with permission. Accessed on 12 September 2025.*

**Figure 3 pharmaceuticals-18-01608-f003:**
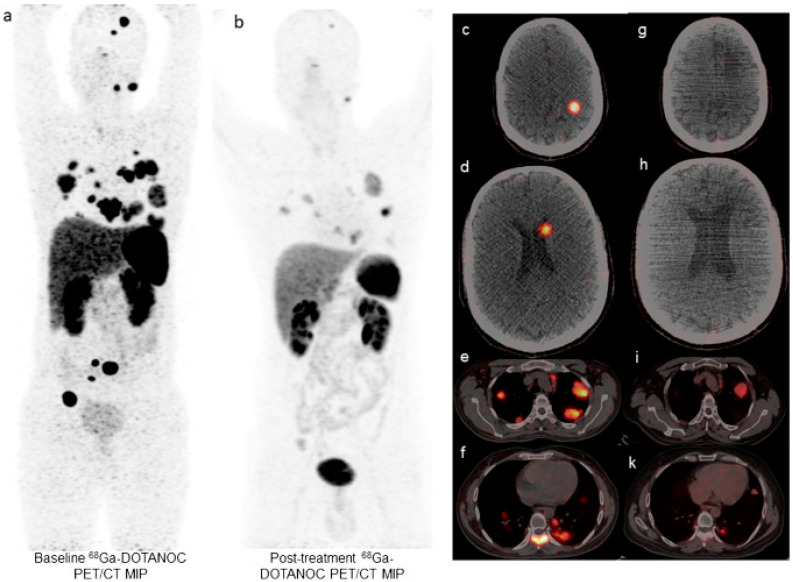
A 41-year-old male patient was diagnosed with a left carotid body tumor in 2006 and underwent excision of the tumor followed by locoregional radiotherapy. The patient received multiple systemic treatments, including ^177^Lu-PRRT and mTOR inhibitor everolimus, but eventually demonstrated disease progression. (**a**) (**c**–**f**) The baseline ^68^Ga-DOTANOC PET/CT scan demonstrates multiple lymph nodes, bilateral lung nodules, skeletal metastases, and brain metastases involvement. (**b**) (**g**–**k**) Restaging PET/CT at two months after the third cycle of ^225^Ac-DOTATATE shows a remarkable reduction in all the corresponding lesions. Note that the greyscale of the MIP and the PET/CT images between two scans are set at the same SUV scale, and the threshold was >2.5. *Figure reproduced from Yadav* MP et al. *Efficacy and safety of ^225^Ac-DOTATATE targeted alpha therapy in metastatic paragangliomas: a pilot study, European Journal of Nuclear Medicine and Molecular Imaging,* [38] *© 2021 Springer Nature, with permission. Accessed on 12 September 2025.*

**Figure 4 pharmaceuticals-18-01608-f004:**
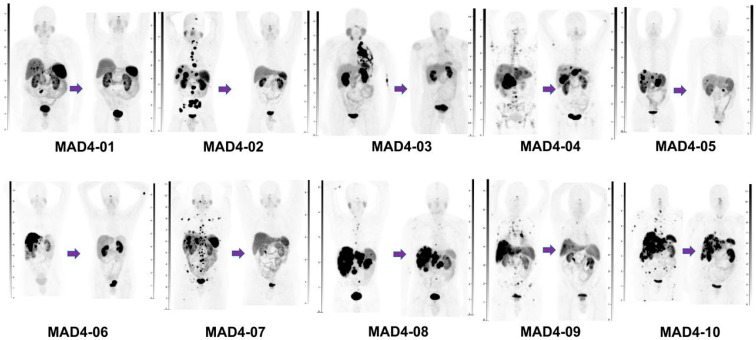
Volume-rendered images of ^68^Ga-DOTATATE PET/CT scans from first 10 subjects enrolled before treatment (**left** side of each panel) and after treatment (**right** side of each panel) with 4 cycles of ^212^Pb-DOTAMTATE at dose of 2.50 MBq/kg for each cycle. *This figure was originally published in JNM. Delpassand* ES et al. *Targeted α-Emitter Therapy with ^212^Pb-DOTAMTATE for the Treatment of Metastatic SSTR-Expressing Neuroendocrine Tumors: First-in-Humans Dose-Escalation Clinical Trial. J. Nucl. Med. 2022;63:1326–1333.* [34] *© SNMMI. Accessed on 12 September 2025.*

**Figure 5 pharmaceuticals-18-01608-f005:**
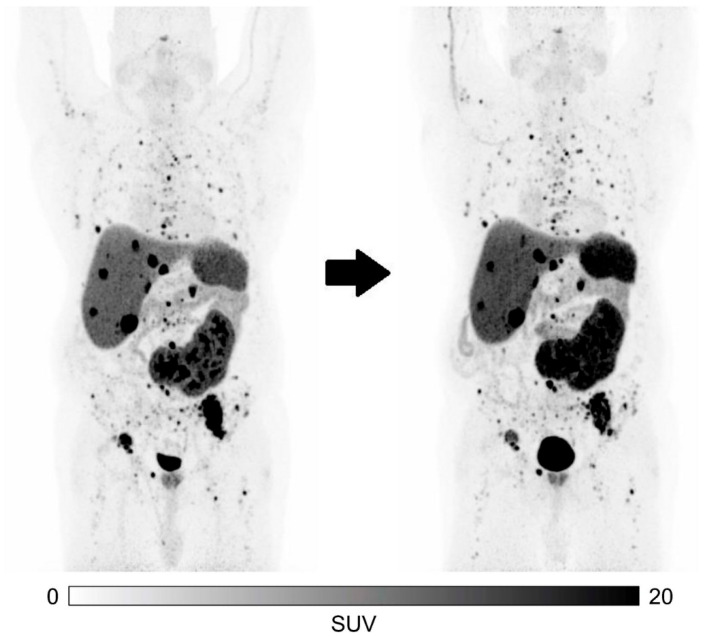
^68^Ga-DOTATATE PET/CT maximum intensity projections (MIP) of a CUP NET G2 patient with multiple hepatic, osseous, and lymphonodal metastases at baseline (**left**) and follow-up (**right**) 3 months after administration of 108 MBq ^212^Pb-VMT-α-NET. Tumor burden, as well as tumor marker CgA, remained stable over time. *Figure reproduced from Michler* E et al., *[^203/212^Pb]Pb-VMT-α-NET as a novel theranostic agent for targeted alpha radiotherapy-first clinical experience, Eur. J. Nucl. Med. Mol. Imaging. 2025 Apr 9 doi: 10.1007/s00259-025-07269-0. Epub ahead of print [39], under CC BY 4.0 (https://creativecommons.org/licenses/by/4.0/). Accessed on 12 September 2025.*

**Table 1 pharmaceuticals-18-01608-t001:** Baseline characteristics of included studies.

Author	Year	Type of Study	Sex/Mean Age (y)		ECOG Score or KPS Score	Site of Primary NET, n	Site of Metastases, n	Treatment Regimen
Ballal et al. [33]	2022	Prospective (n = 91)	M 54 (59.4%) F 37 (40.6%) Mean 54.3 ± 11.6 (range: 25–75)		Median ECOG score = 2 Median KPS score = 60	Foregut: 47 Midgut: 20 Hindgut: 8 Unknown:16	Liver: 88 Lymph nodes: 66 Bone: 25 Lung: 6 Brain: 2 Adrenal glands: 6 Other: 9	[^225^Ac]Ac-DOTATATE cycle every 8 weeks. Mean cumulative activity of 35.52 MBq (range: 21.64–59.47). Median 4 cycles per patient (range: 1–10). Radiosensitizer d0–14.
Ballal et al. [32]	2020	Prospective (n = 32)	M 15 (47%) F 17 (53%) Mean 52 ± 9.2 (range: 35–72)		Baseline ECOG score < 2 Baseline KPS score > 50	Pancreatic: 16 Foregut: 7 Midgut: 3 Hindgut: 1 Unknown: 5	Liver: 29 Lymph nodes: 24 Bone: 12 Other sites: 6	[^225^Ac]Ac-DOTATATE cycle every 8 weeks. Mean cumulative activity of 22.550 kBq ± 9842 (range: 7770–44,400 kBq). Median 3 cycles per patient (range: 1–5).
Delpassand et al. [34]	2022	Prospective (n = 10)	M 6 (50%) F 4 (50%) Range: 39–80		Baseline ECOG score < 2	Pulmonary: 4 Pancreatic: 2 Small bowel: 2 Rectal: 2	NR	[^212^Pb]Pb-DOTAMTATE cycle every 8 weeks. Mean cumulative activity of 791 MBq (range: 681–873). 4 cycles per patient.
Demirci et al. [35]	2023	Retrospective (n = 11)	M 8 (73%) F 3 (27%) Mean 59 ± 11.9 (range: 43–79)		ECOG score not reported Baseline KPS score > 50	Pulmonary: 1 Pancreatic: 3 Non-pancreatic GEP: 1 Paraganglioma: 1 Unknown: 3	Liver: 10 Lymph nodes: 8 Bone: 8 Lung: 4	[^225^Ac]Ac-DOTATATE cycle every 8 weeks. Mean activity of 8.2 MBq ± 0.6 (range: 7.5–10.0 MBq). Median 1 cycle per patient (range: 1–3).
Kratochwil et al. [36]	2014	Retrospective (n = 7)	M 3 (43%) F 4 (57%) Age not reported		Not reported	Pulmonary: 1 Carcinoid: 1 Pancreatic: 4 Unknown: 2	Liver: 7 Bone: 1	[^213^Bi]Bi-DOTATOC cycle every 2 months. Cumulative activity range: 13.3 GBq–20.8 GBq Median 5 cycles per patient (range: 2–5).
Michler et al. [39]	2025	Retrospective (n = 12)	M 9 (75%) F 3 (25%) Mean 71 (range: 60–84)		Baseline ECOG score < 2	Pancreatic: 3 Small bowel: 4 CUP: 3 Rectal: 1 Carcinoid: 1	NR	[^212^Pb]Pb-VMT-α-NET single dose 1.2 MBq/kg
Yadav et al. [38]	2022	Prospective (n = 9)	M 6 (67%) F 3 (33%) Mean 41 ± 10.5 (range: 23–65)	9	Baseline ECOG score < 3 Baseline KPS score > 40	Sympathetic paraganglioma: 3 Parasympathetic paraganglioma: 6	Liver: 2 Lymph nodes: 8 Bone: 6 Lung: 3 Brain: 1 Duodenum: 1	[^225^Ac]Ac-DOTATATE cycle every 8 weeks. Mean cumulative activity of 42.4 MBq ± 27 (range: 15.54–86.6). Median 3 cycles per patient (range: 2–9). Radiosensitizer.
Yang et al. [37]	2024	Prospective (n = 10)	M 7 (70%) F 3 (30%) Mean 47.5 ± 14.1 (range: 26–66)	10	Baseline ECOG score < 3 Baseline KPS score > 40	Pulmonary: 1 Pancreatic: 1 Tonsillar: 1 Paraganglioma: 4 Pheochromocytoma: 3	Liver: 4 Lymph nodes: 9 Bone: 8 Lung: 4 Adrenal glands: 2 Subcutaneous: 1 Muscle: 1	[^225^Ac]Ac-DOTATATE cycle every 8 weeks. Mean cumulative activity of 22.9 MBq ± 9.5 (range: 14.8–44.4 MBq). Median 3 cycles per patient (range: 2–6).

**Table 2 pharmaceuticals-18-01608-t002:** Overview of prior therapies.

Authors	^177^Lu-DOTATATE, n (%)	^90^Y/^177^Lu-DOTATOC, n (%)	^225^Ac-DOTATATE	Surgery, n (%)	Chemotherapy, n (%)	EBRT, n (%)	SSA, n (%)	Targeted Therapies, n (%)	^131^I MIBG, n (%)	TARE/TACE, n (%)	IO, n (%)
Ballal et al. [33]	57 (63%)	NR	NR	21 (23%)	14 (15%)	NR	70 (77%)	10 (11%)	NR	NR	NR
Ballal et al. [32]	32 (100%)	NR	NR	10 (31%)	12 (37%)	NR	28 (87%)	NR	NR	NR	NR
Delpassand et al. [34]	0 (0%)	0 (0%)	NR	NR	NR	0 (0%)	NR	NR	NR	NR	NR
Demirci et al. [35]	10 (91%)	NR	NR	NR	11 (100%)	NR	10 (91%)	NR	2 (18%)	6 (55%)	NR
Kratochwil et al. [36]	0 (0%)	7 (100%)	NR	1 (14%)	3 (43%)	1 (14%)	6 (86%)	1 (14%)	NR	NR	NR
Michler et al. [39]	12 (100%)	NR	6 (50%)	NR	4 (33.3%)	NR	7 (58.3%)	4 (33.3%)	NR	1 (8.3%)	NR
Yadav et al. [38]	7 (78%)	NR	NR	6 (67%)	1 (11%)	5 (55%)	NR	NR	2 (22%)	NR	NR

NR: not reported, EBRT: external beam radiotherapy, SSA: somatostatin analogs, targeted therapies: mammalian target of rapamycin (mTOR) inhibitors and tyrosine kinase inhibitors, TARE/TACE: transarterial radioembolization/transarterial chemoembolization, IO: immunotherapy.

**Table 3 pharmaceuticals-18-01608-t003:** Overview of outcomes.

Author (Year)	Number of Patients, n	Median Follow-Up, Months	Morphologic/Molecular ^ꝉ^ Imaging Response, n (%)	ORR, %	DCR, %	Overall Median PFS, Months	Median OS, Months
Ballal et al. [33]	91	24 (range: 5–41)	CR: 2 (2.5%) PR: 38 (48.1%) SD: 23 (29.1%) PD: 16 (20.3%) NA: 12	50.6%	79.7%	Not reached	Not reached
Ballal et al. [32]	32	8 (range: 2–13)	PR: 15 (62.5%) SD: 9 (37.5%) NA: 8	62.5%	100%	Not reached	Not reached
Delpassand et al. [34]	10	17.4 (range: 9–26)	CR: 1 (10%) PR: 7 (70%) SD: 2 (20%)	80%	100%	Not reached	Not reached
Demirci et al. [35]	11	NR	PR: 4 (44.4%) SD: 4 (44.4%) PD: 1 (11.1%) NA: 2	44.4%	88.9%	12	Not reported
Kratochwil et al. [36]	7	Range: 12–34	CR: 1 (16.7%) PR: 2 (33.3%) SD: 3 (50.0%) NA: 1	50%	100%	Not reached ^‡^	Not reached
Michler et al. [39]	12	3	SD: 8 (100.0%) NA: 4	0%	100%	Not reached	Not reached
Yadav et al. [38]	9	22.5 (range: 18–28)	PR: 4 (50.0%) SD: 3 (37.5%) PD: 1 (12.5%) NA: 1	50%	87.5%	Not reached	Not reached
Yang et al. [37]	10	14 (range: 7–22)	PR: 4 (40%) **^ꝉ^** SD: 4 (40%) **^ꝉ^** PD: 2 (20%) **^ꝉ^**	40%	80%	Not reached	Not reached

CR: complete response, PR: partial response, PD: progressive disease, SD: stable disease, ORR: objective response rate (%), DCR: disease control rate (%), median PFS: median progression-free survival, median OS: median overall survival. ^‡^: not reached taking only hepatic progression into account. **^ꝉ^ Molecular imaging response (PERCIST).**

**Table 4 pharmaceuticals-18-01608-t004:** Overview of ongoing clinical trials.

Author/Sponsor (Year)	Trial No.	Study Type	Radiopharmaceutical	Treatment Regimen	Patient Characteristics	Estimated Enrollment	Outcomes
Yang Zhi, Peking University Cancer Hospital and Institute (2024)	NCT06732505	Phase I	[^225^Ac]Ac-DOTATATE	Dose escalation: Co 1: 90 kBq/kg/cycle Co 2: 120 kBq/kg/cycle Interval of 8 weeks Up to 4 cycles	Inoperable, locally advanced or metastatic, progressive, well differentiated, SSTR+ GEP-NETs (PRRT naïve or with previous PRRT)	36	Safety, optimal IA, ORR, PFS
RayzeBio, ACTION-1 (2022)	NCT05477576	Phase Ib/III	[^225^Ac]Ac-DOTATATE (RYZ101) vs. SoC	Dose escalation: 120 kBq/kg/cycle Interval of 8 weeks Up to 4 cycles	Inoperable, advanced, well-differentiated, SSTR+ GEP-NETs, that have progressed following ^177^Lu-PRRT	288	Safety, optimal IA, PFS
Orano Med, Alphamedix02 (2021)	NCT05153772	Phase II	[^212^Pb]Pb-DOTAMTATE (AlphaMedix)	2.5 MBq/kg/cycle	Inoperable or metastatic, progressive, well differentiated, SSTR+ NETs (PRRT naïve or with previous PRRT)	69	Safety, mPFS, OS, QoL
Markus Puhlmann, Perspective Therapeutics (2023)	NCT05636618	Phase I/IIa	[^212^Pb]Pb-VMT-α-NET	Dose escalation: Co 1: 111 MBq/cycle Co 2: 185 MBq/cycle Co 3: 370 MBq/cycle Co 4: 555 MBq/cycle Interval of 8 weeks Up to 4 cycles	Inoperable or metastatic, well differentiated, SSTR+ NETs (PRRT naïve)	280	Safety, optimal IA, PFS, OS, dosimetry
Joy Zou, Frank Lin, National Cancer Institute (2025)	NCT06479811 NCT06427798	Phase I/II	[^212^Pb]Pb-VMT-α-NET	Dose escalation: No injected activity specified Interval of 8 weeks Up to 4 cycles	Inoperable or metastatic, SSTR+ tumors (GEP-NETs, pheo/pgl, SCLC, RCC, H&N cancers)	120	Safety, optimal IA, ORR, PFS, OS, dosimetry
David Bushnell (2023)	NCT06148636	Phase I	[^212^Pb]Pb-VMT-α-NET	Dose escalation: Injected activity calculated based on kidney dose Interval of 8–10 weeks 2 cycles	Inoperable or metastatic, well differentiated, SSTR+ NETs with previous PRRT treatment	27	Safety, optimal IA, dosimetry

SoC = Standard of Care; ORR = objective response rate; PFS = progression free survival; mPFS = median PFS; OS = overall survival; QoL = Quality of Life; Co = cohort; GEP = gastroenteropancreatic; SSTR = somatostatin receptor; NET = neuroendocrine tumor; pheo = pheochromocytoma; pgl = paraganglioma; SCLC = small cell lung cancer; RCC = renal cell cancer; H&N = head and neck; IA = injected activity.

**Table 5 pharmaceuticals-18-01608-t005:** Overview of adverse events.

Author (Year)	n	Thrombocytopenia, % (n)		Lymphopenia, % (n)		Anemia, % (n)		Kidney Toxicity, % (n)		Liver Toxicity, % (n)		Clinical Adverse Events	Other
Grade of AE Severity		1/2	3/4	1/2	3/4	1/2	3/4	1/2	3/4	1/2	3/4		
Ballal et al. [33]	91	17.6% (16)	1.1% (1)	5.5% (5)	0% (0)	56.0% (51)	0% (0)	12.1% (11)	0% (0)	N/A	0% (0)	Nausea, vomiting, abdominal pain and distention, diarrhea, gastritis, fatigue, musculoskeletal pain, loss of appetite, headache, flushing, myalgia	Malignant ascites and pleural effusion, no tumor lysis syndrome
Ballal et al. [32]	32	6.2% (2)	0% (0)	34.4% (11)	0% (0)	59.3% (19)	0% (0)	3.1% (1)	0% (0)	N/A	0% (0)	Nausea, vomiting, abdominal pain and distention, diarrhea, gastritis, fatigue, musculoskeletal pain, loss of appetite, headache, flushing	No tumor-lysis syndrome
Delpassand et al. [34]	10 *	*	*	*	*	*	*	*	*	*	*	Nausea, fatigue * NB. Toxicity reported for the entire population of 20 patients. In the total cohort, 170 AE were reported of which 49 (29%) were grade 2, 7 (5%) grade 3, and none grade 4.	Transient alopecia, worsening achalasia, CVA, hypoglycemia, dyspnea
Demirci et al. [35]	11	**	0% (0)	**	0% (0)	**	0% (0)	9.1% (1)	0% (0)	NR	0% (0)	NO ** NB. One unspecified grade 2 hematologic toxicity was reported	Ileus
Kratochwil et al. [36]	7	14.3% (1)	0% (0)	14.3% (1)	0% (0)	42.8% (3)	0% (0)	14.3% (1)	0% (0)	NR	NR	NR	Graves’ disease, MDS
Michler et al. [39]	12	0% (0)	0% (0)	62.5% (5)	0% (0)	0% (0)	0% (0)	0% (0)	0% (0)			Nausea, vomiting, fatigue	
Yadav et al. [38]	9	0% (0)	0% (0)	0% (0)	0% (0)	77.8% (7)	0% (0)	0% (0)	0% (0)	0% (0)	0% (0)	Nausea, stomach discomfort, diarrhea	Palpitations, no tumor-lysis syndrome, no life-threatening hypertension
Yang et al. [37]	10	0% (0)	0% (0)	20% (2)	0% (0)	70% (7)	0% (0)	10% (1)	0% (0)	0% (0)	0% (0)	Loss of appetite, nausea and vomiting	No tumor-lysis syndrome

* Toxicity reported for the entire population of 20 patients. In the total cohort, 170 AE were reported of which 49 (29%) were grade 2, 7 (5%) grade 3, and none grade 4. ** One unspecified grade 2 hematologic toxicity was reported.

## Data Availability

No new data were created or analyzed in this study. Data sharing is not applicable to this article.

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
