# Peer review of "Clinical Experience with Targeted Alpha-Emitter Peptide Receptor Radionuclide Therapy (α-PRRT) for Somatostatin Receptor-Positive Neuroendocrine Tumors"

_pharmaceuticals, 2025, doi:10.3390/ph18111608_

Round 1

Reviewer 1 Report

Comments and Suggestions for Authors

The paper entitled “Clinical experience with targeted alpha-emitter peptide receptor radionuclide therapy (α-PRRT) for somatostatin receptor-positive neuroendocrine tumors” by Leupe et al. represents an interesting review summarizing clinical results concerning 225Ac, 213Bi and 212Pb radiopharmaceuticals for targeted alpha therapy of neuroendocrine tumors.

The work is of significant interest to the nuclear medicine and cancer therapy communities and has the potential to make a meaningful contribution to the field. The manuscript is well-written, clearly structured, and provides a comprehensive overview of the current state of clinical research in α-PRRT. The authors successfully highlight both the therapeutic potential and challenges associated with the use of alpha-emitting isotopes.

Furthermore, the discussion section demonstrates a critical understanding of the field, emphasizing the need for larger, controlled trials and long-term follow-up to better define treatment outcomes and optimize patient selection.

Minor comments:

  • Editorial corrections: Table 5: Please add a space between the digits (90, 120) and the unit "kBq"; the abbreviation “Mbq” should be corrected to “MBq” to maintain consistency with standard SI unit formatting.

In summary, this review provides a timely and valuable update on a rapidly evolving therapeutic approach. I fully recommend the manuscript for publication after addressing the very minor editorial revisions noted above.

Congratulations to the authors on an excellent and well-executed review.

Author Response

REVIEWER 1

  • The paper entitled “Clinical experience with targeted alpha-emitter peptide receptor radionuclide therapy (α-PRRT) for somatostatin receptor-positive neuroendocrine tumors” by Leupe et al. represents an interesting review summarizing clinical results concerning 225Ac, 213Bi and 212Pb radiopharmaceuticals for targeted alpha therapy of neuroendocrine tumors. The work is of significant interest to the nuclear medicine and cancer therapy communities and has the potential to make a meaningful contribution to the field. The manuscript is well-written, clearly structured, and provides a comprehensive overview of the current state of clinical research in α-PRRT. The authors successfully highlight both the therapeutic potential and challenges associated with the use of alpha-emitting isotopes. Furthermore, the discussion section demonstrates a critical understanding of the field, emphasizing the need for larger, controlled trials and long-term follow-up to better define treatment outcomes and optimize patient selection.
    • We thank the reviewer for their positive feedback and recognition of the relevance and scope of our review. We appreciate their acknowledgment of the paper’s contribution to summarizing clinical experiences with α-emitter PRRT.
  • Minor comments:
  • Editorial corrections: Table 5: Please add a space between the digits (90, 120) and the unit "kBq"; the abbreviation “Mbq” should be corrected to “MBq” to maintain consistency with standard SI unit formatting.
    • Thank you for noticing this. We added a space between the digits and units, and also revised the other tables to make sure that there is a space between digit and unit in every table. We corrected Mbq to MBq.
  • In summary, this review provides a timely and valuable update on a rapidly evolving therapeutic approach. I fully recommend the manuscript for publication after addressing the very minor editorial revisions noted above.
  • Congratulations to the authors on an excellent and well-executed review.
    • We thank the reviewer once again for the feedback and recognition.

Reviewer 2 Report

Comments and Suggestions for Authors

This is a review of clinical studies of Targeted Alpha-Emitter Peptide Receptor Radionuclide Therapy against neuroendocrine tumors. The review is clearly laid out and easy to read, I have two minor questions.

  1. In Methods you list a number of congresses that you reviewed the abstracts of, I am wondering why Nuclear Medicine focused ones as those of SNMMI and EANM are not included here, seeing as you quote their respective journals quite a lot in the review?
  2. In the discussion you list the different radiopharmaceuticals studied and the experiences from each, but I would have liked to see some text comparing the benefits and drawbacks of each. Maybe not to proclaim a “winner” but give the reader some idea of which is more promising depending on different factors.

Author Response

REVIEWER 2

This is a review of clinical studies of Targeted Alpha-Emitter Peptide Receptor Radionuclide Therapy against neuroendocrine tumors. The review is clearly laid out and easy to read, I have two minor questions.

  1. In Methods you list a number of congresses that you reviewed the abstracts of, I am wondering why Nuclear Medicine focused ones as those of SNMMI and EANM are not included here, seeing as you quote their respective journals quite a lot in the review?
  • We thank the reviewer for this valuable observation. However, late-breaking abstracts of randomized controlled trials (RCTs) in patients, which formed a key focus of our review, are typically presented at oncological congresses, rather than at nuclear medicine–specific meetings. Presentations at EANM and SNMMI are most often readily published in dedicated clinical or nuclear medicine journals. While we also recently attended the EANM congress in 2025 and screened for SNMMI congress materials, no presentations directly relevant to the clinical experience with α-PRRT in neuroendocrine tumors were identified during the review period. Therefore, these congresses were not listed in the Methods section to maintain clarity and focus on the sources that contributed relevant data.
  1. In the discussion you list the different radiopharmaceuticals studied and the experiences from each, but I would have liked to see some text comparing the benefits and drawbacks of each. Maybe not to proclaim a “winner” but give the reader some idea of which is more promising depending on different factors.
  • Thank you for this suggestion. This is indeed relevant to our work. We discussed this in depth and added the following text to the discussion:

“Among the α-emitters investigated for peptide receptor radionuclide therapy (α-PRRT), actinium-225 and lead-212 have emerged as the most promising candidates due to their favorable decay characteristics and compatibility with peptide-based targeting vectors.

Actinium-225 is currently the most clinically advanced α-emitter. With a half-life of 9.9 days, it emits four α-particles per decay chain, enabling highly potent tumor irradiation. Its labeling chemistry with DOTA-based peptides allows stable binding and effective systemic delivery. Clinical studies using [²²⁵Ac]Ac-DOTATATE have demonstrated strong therapeutic efficacy with manageable toxicity. However, a key safety concern is the recoil-induced release of radioactive daughters. The α-decay of ²²⁵Ac generates radionuclides such as francium-221 and bismuth-213 and bismuth-213, which can detach from the chelator and redistribute to non-target tissues. Free francium-221 tends to accumulate in kidneys, salivary glands, and intestines, while bismuth-213 shows preferential retention in kidneys and liver. The magnitude of potential off-target toxicity is vector-dependent and warrants careful consideration in radiopharmaceutical design.

Lead-212, with a 10.6-hour half-life, decays in vivo to Bismuth-212, thereby extending the effective therapeutic window for α-particle emission. Both local and centralized production of 212Pb-radiopharmaceuticals are feasible. Preclinical and early clinical evaluations of [²¹²Pb]Pb-DOTAMTATE and related compounds have shown encouraging results, demonstrating the practical feasibility and scalability of this isotope. Furthermore, imaging surrogates such as lead-203 allow for dosimetry assessments and theranostic applications.

In contrast, bismuth-213 has a short half-life of 45 minutes, leading to rapid and intense dose deposition over a brief period. While this property can be advantageous for highly localized therapies, it limits the feasibility of systemic applications unless used with a rapidly clearing vector that minimizes renal retention. The short half-life also necessitates on-site production and quality control of ²¹³Bi-labeled radiopharmaceuticals, posing significant logistical challenges. Although high-activity ²²⁵Ac/²¹³Bi generator systems have been developed (Morgenstern, A.; Apostolidis, C.; Bruchertseifer, F. Supply and Clinical Application of Actinium-. Semin. Nucl. Med. 2020, 50, 119–123. AND McDevitt, M.R.; Finn, R.D.; Sgouros, G.; Ma, D.; Scheinberg, D.A. An 225Ac/213Bi generator system for therapeutic clinical applications:Construction and operation. Appl. Radiat. Isot. 1999, 50, 895–904.), the primary bottleneck remains the limited global availability of the parent isotope actinium-225. However, ongoing efforts to expand ²²⁵Ac production are expected to alleviate these constraints in the coming years, potentially paving the way for broader clinical adoption of ²¹³Bi-based radiopharmaceuticals (Ahenkorah, S.; Cassells, I.; Deroose, C.M.; Cardinaels, T.; Burgoyne, A.R.; Bormans, G.; Ooms, M.; Cleeren, F. Bismuth-213 for Targeted Radionuclide Therapy: From Atom to Bedside. Pharmaceutics 2021, 13, 599. https://doi.org/10.3390/pharmaceutics13050599). Global ²²⁵Ac production by 2032 is estimated at well above 25 TBq per year, which is sufficient to produce at least 2 million patient doses, assuming average treatment needs (average 30–36 MBq per patient) (Zimmermann R. Is Actinium Really Happening? J Nucl Med. 2023 Oct;64(10):1516-1518. doi: 10.2967/jnumed.123.265907. Epub 2023 Aug 17. PMID: 37591546.).

Astatine-211 possesses nearly ideal properties for α-PRRT. Its 7.2-hour half-life, pure α-emission, and absence of long-lived α- or high-energy γ-emitting daughters make it particularly suitable for small peptides and rapidly clearing biomolecules. However, the chemical instability (in vivo deastatination) of many ²¹¹At-labeled radiopharmaceuticals remains a key challenge that must be addressed to fully harness the therapeutic potential of this isotope. Despite its favorable radiophysical and dosimetric profile, clinical translation is constrained by the restricted availability of suitable cyclotrons and radiochemistry facilities capable of producing ²¹¹At-radiopharmaceuticals at clinically relevant scales. Moreover, although astatine-211 itself decays relatively quickly, a small fraction of decays produces bismuth-207, with a long half life of 31.6 years, which necessitates long-term storage or regulated disposal of waste. While the absolute activity of bismuth-207 is low, its presence imposes waste-management considerations beyond those associated with shorter-lived α-emitters such as lead-212 or bismuth-213. To date, no clinical α-PRRT studies in patients have been reported using astatine-211.”
